# Brain dynamics for confidence-weighted learning

**Florent Meyniel** ⓘ *

Cognitive Neuroimaging Unit, NeuroSpin center, Institute for Life Sciences Frédéric Joliot, Fundamental Research Division, Commissariat à l'Energie Atomique et aux énergies alternatives, INSERM, Université Paris-Sud, Université Paris-Saclay, Gif-sur-Yvette, France

\* florent.meyniel@cea.fr

## Abstract

Learning in a changing, uncertain environment is a difficult problem. A popular solution is to predict future observations and then use surprising outcomes to update those predictions. However, humans also have a sense of confidence that characterizes the precision of their predictions. Bayesian models use a confidence-weighting principle to regulate learning: for a given surprise, the update is smaller when the confidence about the prediction was higher. Prior behavioral evidence indicates that human learning adheres to this confidence-weighting principle. Here, we explored the human brain dynamics sub-tending the confidence-weighting of learning using magneto-encephalography (MEG). During our volatile probability learning task, subjects' confidence reports conformed with Bayesian inference. MEG revealed several stimulus-evoked brain responses whose amplitude reflected surprise, and some of them were further shaped by confidence: surprise amplified the stimulus-evoked response whereas confidence dampened it. Confidence about predictions also modulated several aspects of the brain state: pupil-linked arousal and beta-range (15–30 Hz) oscillations. The brain state in turn modulated specific stimulus-evoked surprise responses following the confidence-weighting principle. Our results thus indicate that there exist, in the human brain, signals reflecting surprise that are dampened by confidence in a way that is appropriate for learning according to Bayesian inference. They also suggest a mechanism for confidence-weighted learning: confidence about predictions would modulate intrinsic properties of the brain state to amplify or dampen surprise responses evoked by discrepant observations.

**Data Availability Statement:** The full, raw data set is available at https://osf.io/37e68.

**Funding:** This work was funded by College de France (to FM) and the French National Research Agency (to FM, ANR-18-CE37-0010 "CONFI-LEARN", https://anr.fr/en/). The funders had no

## Author summary

Learning in a changing and uncertain world is difficult. In this context, facing a discrepancy between my current belief and new observations may reflect random fluctuations (e.g. my commute train is unexpectedly late, but it happens sometimes), if so, I should ignore this discrepancy and not change erratically my belief. However, this discrepancy could also denote a profound change (e.g. the train company changed and is less reliable), in this case, I should promptly revise my current belief. Human learning is adaptive: we

role in study design, data collection and analysis, decision to publish, or preparation of the manuscript.

**Competing interests:** The authors have declared that no competing interests exist.

change how much we learn from new observations, in particular, we promote flexibility when facing profound changes. A mathematical analysis of the problem shows that we should increase flexibility when the confidence about our current belief is low, which occurs when a change is suspected. Here, I show that human learners entertain rational confidence levels during the learning of changing probabilities. This confidence modulates intrinsic properties of the brain state (oscillatory activity and neuromodulation) which in turn amplifies or reduces, depending on whether confidence is low or high, the neural responses to discrepant observations. This confidence-weighting mechanism could underpin adaptive learning.

## Introduction

Popular learning algorithms like predictive coding [1–4] and the delta rule [5,6] posit that expectations (or equivalently here, predictions) play a key role in learning from a sequence of observations. Those algorithms, in their simplest form, consist in updating the quantity that is learned, in proportion to the prediction error, which is the difference between the prediction and the actual observation. This solution is simple and yet efficient even in changing and uncertain environments [6–8]. Other, more sophisticated algorithms exist [9–14], they formalize the discrepancy between predictions and observations differently (prediction error, improbability, surprise) but they all have in common that this discrepancy is the driving force of learning.

Recently, another aspect of human learning has been put forward: learning is accompanied by a sense of confidence about predictions. Interestingly, this sense of confidence follows, at least in part, the optimal principles of probabilistic inference; in that sense, it is rational [10,15,16]. Here, we embrace the proposal that, in a learning context, the sense of confidence plays a functional role [10,17–19]: it regulates learning according to the confidence-weighting principle. This principle is prescribed by the optimal rules of probabilistic inference and is exhibited in particular by Bayesian models, which obey those rules. In such models, the update is not only guided by discrepant observations, but also regulated by confidence about predictions: for a given discrepancy, the update is smaller when the confidence associated with the prediction was larger. In other words, in a learning context, confidence should set the balance between predictions and new data in order to update the current belief. This confidence-weighting principle is not specific to learning, it is generally applicable whenever several sources of information must be combined [18,20–24]. Confidence-weighting is also related to other notions discussed at the end of this article (selective attention [18,25], the weight of evidence [24], the precision of predictions [12,17,26]).

We propose to use optimal Bayesian models as a benchmark to formalize, at a computational level, the learning process. Our task presents long sequences composed of two stimuli (A or B); we formalize several latent variables of learning as follows (see Eqs 3–5 in Methods).

- The **prediction** about the next stimulus corresponds to the estimated probability of the stimulus, p(A). The Bayesian model actually estimates (or learns, infers; they are synonyms here) a full distribution indicating the likelihood of any possible value of p(A) given the previous observations; the prediction is thus more specifically the expected value of p(A) (i.e. the mean of the distribution). Note that p(B) can be deduced from p(A) as p(B) = 1-p(A); one can thus focus arbitrarily on p(A) rather than p(B) without loss of generality.

- We formalize the discrepancy between predictions and observations as the information-theoretic measure of **surprise** [27–29]. Technically, surprise is the estimated negative log probability of the observed stimulus; it is similar (but not identical) to the unsigned prediction error [3]: one increases non-linearly as a function of the other.

- Last, we formalize the **confidence about a prediction**, as the log-precision (i.e. log(1/variance)) of the full distribution over the prediction p(A) [15,16,30,31].

Previous studies reported conflicting results about the existence of a confidence-weighting mechanism in the brain [19,32,33]. Here, we used a probability learning task previously developed, in which the participants' reports of probability and the associated confidence levels are compatible with optimal Bayesian inference. Our goal is two-fold. Numerous studies reported the existence of surprise signals in the brain, i.e. neural responses that are more vigorous for unexpected stimuli [29,34–42]. We will test whether surprise responses show an additional effect of confidence as prescribed by the confidence-weighting principle, making them suitable for close-to-optimal updates. Second, previous studies suggested that stimulus-evoked responses are shaped by intrinsic properties of brain networks (which we will term *brain state* for brevity). Those properties are changing endogenously (rather than being merely evoked by a stimulus [43–47]), they are functional (they regulate information processing [48–53]) and at least in part under top-down control (subtending for instance selective attention [54–56]). Those properties are characterized by specific synchronization of oscillatory activity particularly in low frequency bands [48,49,51,52,57] and controlled by neuromodulators [43–45,53,55,56]. We will test whether confidence, which changes depending on the sequence of stimuli following the Bayes optimal model, modulates the brain state by inspecting spectral properties of brain activity and the neuromodulatory state reflected by non-luminance related changes in pupil size. Last, we will test whether those intrinsic aspects of the brain state shape surprise signals in the brain, thereby offering a possible neural mechanism for confidence-weighted learning.

## Results

### Studying confidence in a learning context with a Bayes optimal model

We adopted a probability learning task (**Fig 1A**) that is amenable to (optimal) Bayesian modeling, and in which the human inference is well accounted for by a Bayesian model. Subjects were presented with auditory sequences made of two tones (say A and B), presented randomly according to predefined transition probabilities between successive tones: p(A|A) and p(B|B) (we motivate the use of transition probabilities below). Note that p(B|A) and p(A|B) are deduced from the two other quantities as 1-p(A|A) and 1-p(B|B). Those transition probabilities changed abruptly and unpredictably in the course of the experiment, at "change points". The generative model of the task therefore has three levels, organized hierarchically: 1) the sequence of tones, 2) the transition probabilities governing the tones and 3) the probability of change point, which was fixed across trials. We used Bayesian inference to invert the generative process of the task (**Fig 1B**) and estimate, at any given trial, the transition probabilities currently generating the tones, given previous observations and knowledge of the actual task structure. This inference returns full posterior distributions for the transition probabilities p(A|A) and p(B|B), see **Fig 1C**.

Subjects, as the Bayes-optimal model, were fully informed about the task structure; the only difference is that they were not given the numeric value of the frequency of change points, but the qualitative indication that they are rare. We probed their inference *occasionally* (i.e. once in a while) by asking them to report the probability of the next tone (question #1), and then

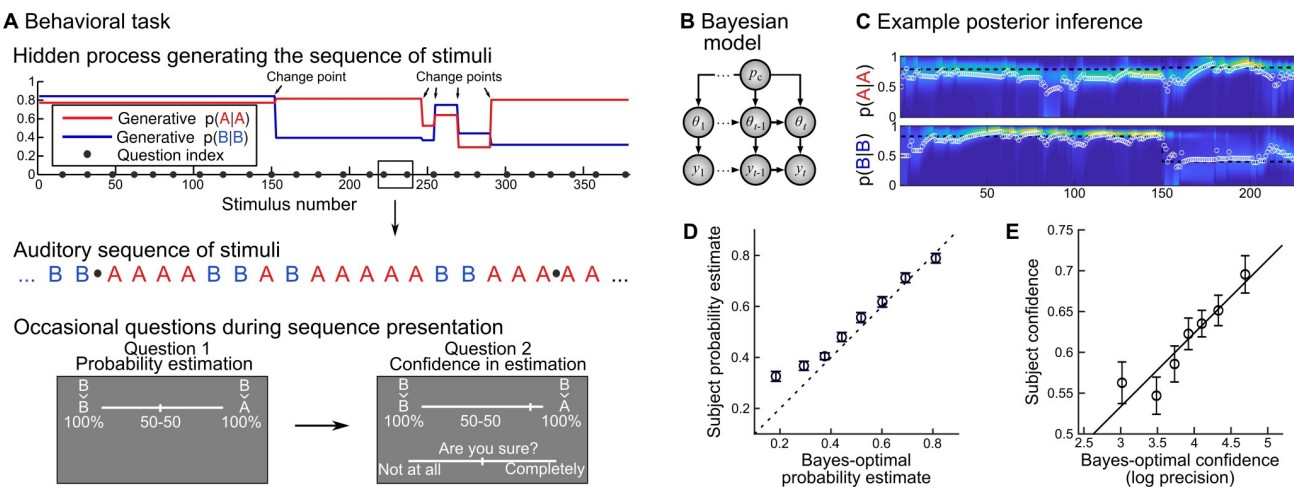

**Fig 1. Rational confidence levels during a probability learning task.** (A) Subjects listened to sequences of two tones (denoted A and B) generated randomly following (order 1) transition probabilities that changed abruptly at unpredictable moments. Subjects knew about this generative process and the tones were perceived without ambiguity. Occasionally subjects were asked to report the probability of the next tone, which amounts to reporting the transition probability relevant at the moment of the question, and then their confidence about this probability estimate. They used continuous sliders for both reports. (B) We designed a Bayes-optimal, hierarchical model that corresponds to the generative process used in the experiment: given current and past observations *y*s, it estimates the momentary transition probabilities $\theta_t$, knowing that there is a fixed probability $p_c$ (= 1/75) of a change point at every trial. (C) The Bayes-optimal, hierarchical model returns a posterior, time-varying, two-dimensional distribution, each dimension corresponding to a transition probability. This distribution evolves as a function of the observations received; the heat-map shows the evolving distribution, color-coded and stacked horizontally (a vertical slice corresponds to the distribution at a given moment). We formalized the answers to question 1 (probability report) and 2 (confidence) as the mean of the distribution (white circle) and its log-precision, respectively. (D-E) Subjects' reports are plotted against equally-filled bins of Bayes-optimal values, as mean ± s.e.m.

their confidence about this estimate (question #2). In the Bayes-optimal model, the answer to question #1 corresponds to the mean of the posterior distribution of the relevant transition probability (the one that corresponds to the tone presented on the previous trial), and we formalized the answer to question #2 (confidence) as the log-precision of this distribution (see Methods) because when precision is high, the posterior evidence is highly concentrated around the mean, which gives credence to this value (**Fig 1C**).

Below, we leverage several features of the task. First, the Bayes-optimal model provides an account of the subjects' probability estimates and confidence during the question trials (see "Behavioral results"), which licenses the use of this good-enough model to study the brain mechanisms of inference during the no question trials. The no question trials are advantageously numerous and unperturbed by motor artifacts or other processes related to answering questions. Second, change points induce frequent trial-to-trial upward and downward variations in confidence in both the optimal Bayesian model and subjects, which facilitates the exploration of the neural correlates of confidence. In the absence of change points, confidence would essentially increase steadily with the trial number (because estimates get more precise with more data points), thereby hindering the quest of correlates of confidence. Third, the use of transition probabilities further increases trial-to-trial variations in confidence: when p(A|A) and p(B|B) are associated with markedly different levels of confidence, then confidence about the prediction greatly changes depending on whether the previous observation was A or B (see **S1 Fig** for an example).

## Behavioral results

Subjects' probability estimates were highly correlated with the Bayes-optimal model (**Fig 1D**): Pearson $\rho = 0.62 \pm 0.04$ s.e.m., 95% CI = [0.55 0.70], Cohen d = 3.42, $t_{23} = 16.75$, p = 2.2 $10^{-14}$.

The same was true for confidence reports, although to a lesser extend (**Fig 1E**): $\rho = 0.23 \pm 0.04$ s.e.m., 95% CI = [0.15 0.30], Cohen d = 1.26, $t_{23} = 6.19$, p = 2.6 $10^{-6}$. Beyond mere correlation, it is also instructive to test for qualitative features of the Bayes optimal model. Several features of subjects' confidence conformed to the Bayesian model, which replicates the results of a previous study [15].

Confidence was not linearly related to the probability estimate in the Bayes-optimal model (Pearson $\rho = 0.04 \pm 0.03$ s.e.m., 95% CI = [-0.03 0.11], Cohen d = 0.24, $t_{23} = 1.19$, p = 0.25), nor in subjects (Pearson $\rho = -0.004 \pm 0.036$ s.e.m., 95% CI = [-0.08 0.07], Cohen d = -0.02, $t_{23} = -0.12$, p = 0.90). However, there was a U-shape relationship: confidence is typically higher when the estimated probability $p$ approaches 0 or 1 (i.e. when the next observation is more predictable), which was captured, in a multiple linear regression of confidence including both $p$ and $p^2$, as a significant weight on $p^2$, in the Bayes-optimal model ($\beta = 0.64 \pm 0.04$ s.e.m., 95% CI = [0.55 0.74], Cohen d = 2.94, $t_{23} = 14.4$, p = 5.5 $10^{-13}$) and in subjects ($\beta = 0.21 \pm 0.03$ s.e.m., 95% CI = [0.15 0.26], Cohen d = 1.58, $t_{23} = 7.76$, p = 7.2 $10^{-8}$).

In the optimal model, confidence also tends to be lower when the previous observation was more surprising. A linear regression of confidence captured this effect with a negative weight of surprise ($\beta = -0.22 \pm 0.01$ s.e.m., 95% CI = [-0.24 -0.21], Cohen d = -6.61, $t_{23} = -32.39$, p = 1.1 $10^{-20}$). We could not test whether subjects were also less confident when more surprised because we do not have access to their prediction on the previous trial (and thus, the ensuing surprise) due to the occasional nature of behavioral reports in our task. However, we can test for a relationship between their reported confidence and the Bayes-optimal surprise on the previous observation, which was also negative ($\beta = -0.04 \pm 0.01$ s.e.m., 95% CI = [-0.06 -0.02], Cohen d = -1.10, $t_{23} = -5.41$, p = 1.7 $10^{-5}$).

The Bayes-optimal confidence also tends to increase as more observations are accumulated within a stable period (i.e. since the last change point). This increase should be linear with the log number of observations (a result that can be derived analytically), and indeed captured as a linear effect in the model ($\beta = 0.16 \pm 0.01$ s.e.m., 95% CI = [0.14 0.19], Cohen d = 2.68, $t_{23} = 13.11$, p = 3.7 $10^{-12}$) and subjects ($\beta = 0.02 \pm 0.01$ s.e.m., CI = [0.01 0.03], Cohen d = 0.64, $t_{23} = 3.13$, p = 4.7 $10^{-3}$).

Those three features of confidence (effects of $p^2$, surprise, log-number of observations) could constitute a heuristic estimation of confidence approaching optimality. However, subjects' confidence was not reducible to those effects. We estimated a multiple linear regression of the subject's confidence including many potential predictors: the Bayes-optimal estimate of $p$, $p^2$, $\log(p)$ and $\log(1-p)$ (in order to capture non linear effects), the same quantities but with the subject's probability ($p_s$, $p_s^2$, $\log(p_s)$ and $\log(1-p_s)$), the optimal estimate of last surprise, the optimal estimate of the last prediction error (as in linear predictive coding [3]), and the log number of observations since the last change point. Despite those eleven predictors, the residual subject's confidence still covaried significantly with Bayes-optimal confidence ($\beta = 0.012 \pm 0.003$ s.e.m., 95% CI = [0.005 0.019], Cohen d = 0.70, $t_{23} = 3.44$, p = 2.2 $10^{-3}$), ruling out that it is reducible to those sophisticated heuristics.

## Evidence for a confidence-weighting of surprise in evoked responses

Many brain responses evoked by stimuli are known to be modulated by surprise: they are more vigorous for unexpected stimuli. Here, we tested whether we could identify such surprise responses and whether they would show additional effects of confidence following the confidence-weighting principle, i.e. dampened responses for higher confidence. To this end, we estimated a multiple regression model, systematically and independently for all sensors and peri-stimulus times, from -0.1 to 0.8 s. Note that the inter-stimulus interval is 1.4 s, such that this time window only contains

the current stimulus. We included three predictors: the stimulus identity (coded as a binary variable), the Bayes-optimal surprise to the current stimulus and the Bayes-optimal confidence about the prediction of the current stimulus identity. This analysis yielded significant results for the three predictors (cluster-forming p<0.001, cluster-level p<0.05, two-tailed, n = 1,000).

We also performed a more conservative analysis, yielding very similar results. Some aspects of confidence being related to the probability estimate itself (see behavioral section above), the effect of confidence we found could be confounded with some aspects of the prediction itself. In order to rule out this possibility, we estimated another regression model, after replacing the Bayes-optimal confidence with the *residual confidence*. From now on, *residual confidence* will denote what remains in the Bayes-optimal confidence after regressing out several effects related to the probability estimate (Bayes-optimal p, $p^2$, log(p), log(1-p); using the square and logs enables to capture non-linear effects). For instance, unlike confidence and by construction, residual confidence does not increase when the identity of the next observation is more certain (p closer to 0 or 1). **Fig 2A** shows the result of the analysis with residual confidence, and various significant effects (cluster-forming p<0.001, cluster-level p<0.05, two-tailed, n = 1,000): the stimulus identity (105–160 ms) co-occurring with a first surprise response (80–155 ms) followed by a second one (160–290 ms) and a late, prolonged one (360–715 ms). Importantly, an effect of confidence (155–230 ms) overlapped in space and time with the effect of surprise. The activity time course in the confidence cluster (**Fig 2B**), provided for illustrative purpose, shows that the response around 200 ms, characterized by an inward field (negative sign), is more vigorous for unexpected stimuli, and dampened for higher confidence, akin to the confidence-weighted surprise signal predicted by Bayesian learning. Source reconstruction within 155–230 ms (**Fig 2C**) revealed that the negative effect of confidence and the positive effect of surprise on brain activity showed some anatomical overlap.

## Confidence-weighting of surprise in evoked responses: control analyses

We further tested the robustness of the confidence effect found within 155–230 ms in the significant sensors shown in **Fig 2B**. First, the regression model did not include an interaction between surprise and confidence, because their effects on model update is theoretically mostly additive, such that looking for a putative update signal corresponds to estimating a linear regression model with surprise and confidence, and then looking for an overlap of those two effects (see Methods for details). Their interaction, when included in the previous linear regression model, was not significant ($\beta$ = 3.3 $10^{-16}$ ± 1.1 $10^{-15}$ s.e.m., 95% CI = [-0.20 0.26] $10^{-14}$, Cohen's d = 0.06, $t_{23}$ = 0.29, p = 0.77) and left the results unchanged. Second, the temporal profiles of Bayes-optimal confidence share commonalities across subjects, for instance, it is low at the beginning of a session, and then waxes and wanes multiple times (see **S1 Fig**). One concern is that those temporal characteristics may drive, for spurious reasons, the correlation between confidence and the MEG signal. To rule out this possibility, we shuffled the time-courses of (residual Bayes-optimal) confidence with respect to MEG data across subjects to estimate a null distribution for the correlation between confidence and MEG signal, and a Z-statistics. This approach controls for the temporal characteristics of Bayes-optimal confidence shared across all sequences presented to subjects, it is thus very conservative; yet the correlation between confidence and MEG signal remained significant (z = 0.95 ± 0.19 s.e.m., 95% CI = [0.55 1.35], Cohen's d = 1.00, $t_{23}$ = 4.92, p = 5.7 $10^{-5}$).

## Modulation of the brain state by confidence: Low frequency oscillations

Next, we examined whether Bayes-optimal confidence would correlate with spectral components of the MEG signal. Power in low frequencies (<40 Hz) typically characterizes the state of

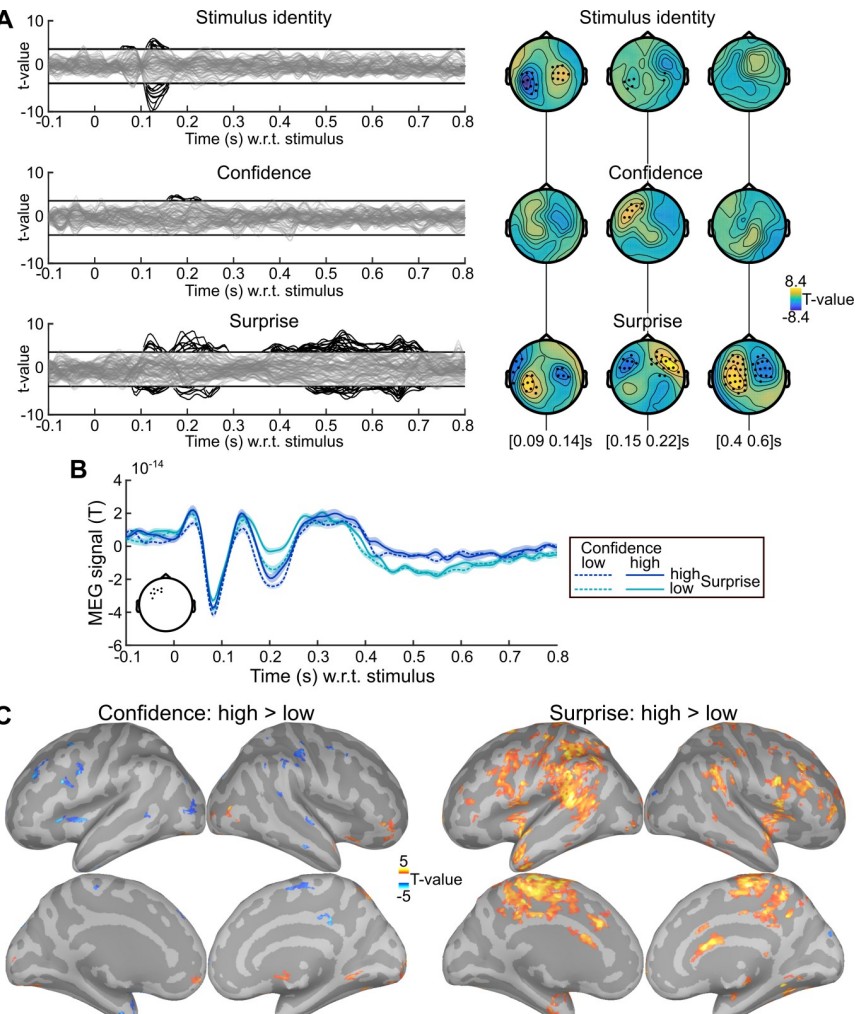

**Fig 2. Evidence for surprise and confidence-weighted surprise in evoked activity.** (A) The stimulus-locked evoked response was analyzed with mass-univariate regression. The value at each sensor and peri-stimulus time was regressed, across trials, using a multiple linear model with three predictors: the stimulus identity, the (residual) Bayes-optimal confidence about the probability of the current stimulus and the Bayes-optimal surprise (i.e. negative log likelihood). The time series show the significance, as t-values, of regression coefficients, with one line per sensor; topographies show the t-value of effect averaged within three time windows. Highlighted time points and sensors survive correction for multiple comparisons across peri-stimulus times and sensors (cluster-forming p<0.001 two-tailed, cluster-level p<0.05 two-tailed, n = 1,000). (B) Time series of MEG signal (mean ± s.e.m. estimated after subtracting the subject-mean) in the cluster of sensors (see inset) showing a significant effect of Bayes-optimal confidence. Trials were sorted into high and low surprise (median split) and high and low confidence (median split); this conjunction of splits resulted in 333.1 to 334.5 trials on average across subjects in each of the four conditions. The response around 200 ms is more extreme for higher surprise and less extreme for higher confidence. (C) Source reconstruction was computed within 155–230 ms for trials with high and low confidence (median split), high and low surprise (median split). The significance of the paired difference between high and low levels is shown separately for confidence and surprise (threshold: p<0.05, two-tailed, minimum cluster size: 50 vertices, total vertices: 306,716). Increased signal for higher surprise and lower signal for higher confidence overlapped in several brain regions.

large networks [44,46,47,50,52,54,57]. Depending on this state, those networks could respond differently to incoming stimuli, thereby implementing a confidence-weighting mechanism of surprise (see Discussion). The signal was decomposed across peri-stimulus times and frequencies (6–40 Hz) and analyzed from -0.5 to 0.9 s. We used the same regression model as used for evoked response, with stimulus identity, Bayes-optimal surprise and residual Bayes-optimal

confidence. We found significant effects (cluster-forming $p<0.001$ and cluster-level $p<0.01$ two-tailed, corrected for multiple comparisons, n = 2,000) of surprise and confidence which were all positive: the higher the confidence (or surprise), the higher the power. The surprise effect, given its topography, timing and low frequency simply reflected the evoked responses identified in **Fig 2A**. In contrast, the confidence effect was much more extended in terms of time, frequency and sensor. For simplicity, we grouped significant clusters in pre and post-stimulus clusters, and further sub-divided the latter in alpha and beta range, with respect to 12 Hz (**Fig 3A**). Beta band activity was particularly increased in the prefrontal cortex when confidence was higher (**Fig 3B**). Note that the confidence effect was present even before the stimulus, see for illustration the time courses (**Fig 3C**) and topography (**Fig 3D**) of the confidence effect in the 15–25 Hz band.

## Low frequency oscillations: Control analyses

We further tested the robustness of those confidence effects in those three clusters. All clusters passed the tests, but showed that the effect was most robust in the post-stimulus, beta-band cluster; therefore, from now on we focus on this one (unless otherwise specified). First, we controlled that the correlation we found was not driven by the general temporal profile of Bayes-optimal confidence in the course of experimental sessions, using the same permutation procedure as described for the evoked responses. The effect remained significant ($z = 1.16 \pm 0.16$ s.e. m., 95% CI = [0.84 1.49], Cohen's d = 1.50, $t_{23} = 7.35$, $p = 1.8 \ 10^{-7}$). We also tested whether the effect survived the inclusion of confidence from the previous trial in the regression model, which was true: $\beta = 2.5 \ 10^{-26} \pm 4.3 \ 10^{-27}$ s.e.m., 95% CI = [1.66 3.44] $10^{-26}$, Cohen's d = 1.21, $t_{23} = 5.93$, $p = 4.8 \ 10^{-6}$; this test is particularly relevant for the pre-stimulus cluster ($\beta = 6.8 \ 10^{-26} \pm 1.8 \ 10^{-26}$ s.e.m., 95% CI = [3.1 10.4] $10^{-26}$, Cohen's d = 0.79, $t_{23} = 3.85$, $p = 8.2 \ 10^{-4}$), denoting a preparation of brain networks to the upcoming stimulus based on the confidence held specifically in the prediction. This effect suggests that power can change from one trial to the next, following changes in confidence. We tested this possibility by considering pairs of consecutive trials, sorting them into low and high confidence (median split) on the current trial and further sorting them into high vs. low confidence (median split) on the next trial. High-to-low and low-to-high transitions in Bayes-optimal confidence indeed corresponded respectively to decreases and increases in power from one trial to the next, see **S2 Fig**.

If some spectral properties of neural signals correlate with the (Bayes-optimal) confidence about the prediction even before the presentation of the next stimulus, then they may be predictive of the confidence reported by subjects when the stimulus is actually replaced by a question. Note that a similar analysis was not possible with stimulus-evoked responses (Fig 2) because the stimulus is actually replaced by the question. We trained a ridge regression model to predict the subject's confidence report based on the pre-stimulus power (average between -500 and 0 ms with respect to the omitted stimulus) in frequencies ranging from 6 to 40 Hz, and evaluated its predictive accuracy in a cross-validated manner (see Methods). Out-of-sample predictions co-varied weakly but significantly with subject's confidence (Pearson $\rho = 0.06 \pm 0.02$ s.e.m., 95% CI = [0.019 0.101], Cohen's d = 0.50, $t_{23} = 2.43$, $p = 0.023$, **Fig 3E**). To explore the frequencies contributing to this predictive power, we estimated the ridge regression weights on the full dataset of each subject (see **Fig 3F**). Those weights must be interpreted with caution because they depend on the covariance of the data, and may be different across subjects; nevertheless they suggest a positive effect of (low) beta range power (15–20 Hz) similar to the effect of Bayes-optimal confidence during the no-question trials. However, power in the alpha range (around 10 Hz) was negatively related to subjec's confidence, unlike the effect

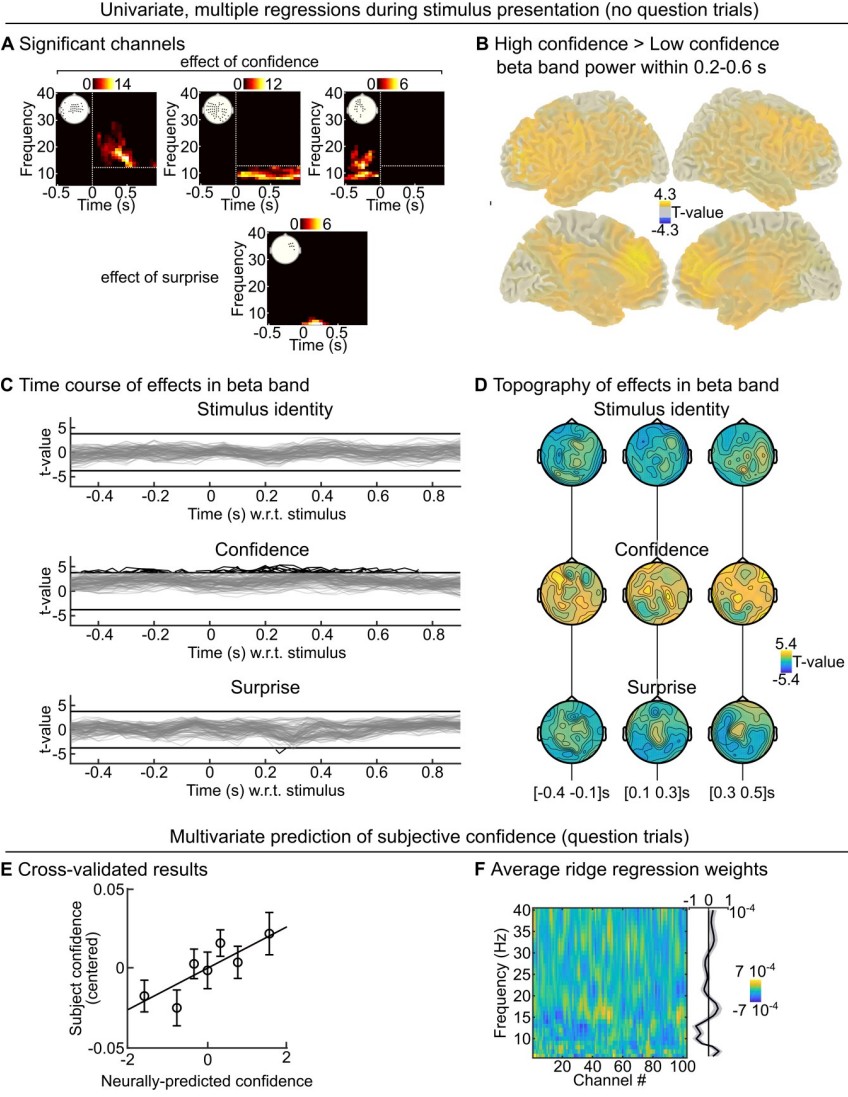

**Fig 3. Beta-range oscillations reflect optimal and subjective confidence.** (A) The peri-stimulus power was analyzed with mass-univariate regression. The value at each sensor, peri-stimulus time and frequency was regressed, across trials, using a multiple linear model with three predictors: the stimulus identity, the (residual) Bayes-optimal confidence about the probability of the current stimulus and the Bayes-optimal surprise. We used correction for multiple comparisons across peri-stimulus times, sensors and frequencies (cluster-forming p<0.001, cluster-level p<0.01 two-tailed, n = 2,000). Several clusters were found for Bayes-optimal confidence, which were regrouped for clarity as: post-stimulus effect in beta band (>12 Hz); post-stimulus effect in the alpha band (≤12 Hz) and pre-stimulus effect in alpha-beta band. For each cluster, the inset shows the significant sensors and the heat-map shows the number of significant sensors for each time-frequency pair. (B) Source reconstruction of power was computed within 0.2–0.6 s and 15±3 Hz for trials with high and low confidence (median split). The significance of the paired difference between high and low confidence is shown (threshold: p<0.05, two-tailed). (C-D) For illustration, we show the significance levels (as t-value) of each predictor variable in the 15–25 Hz band, as time courses (C, one line per sensor; the threshold correspond to p<0.001 two-tailed) and topographies (D). (E-F) During question trials, the stimulus was replaced by a question, asking the subject for her prediction about the stimulus identify. We trained a multivariate ridge regression model in a cross-validated manner to predict, based on the pre-stimulus power (average within -500 to 0 ms relative to question) across frequencies, the confidence report of the subject. D shows the cross-validated prediction accuracy (mean ± s.e.m.) using equally-filled bins of data, and E shows the (non cross-validated) regression weights for each frequency and channels (together with the average across channels).

of optimal confidence on power during no-question trials, suggesting the existence of multiple processings behind subjective confidence.

## Modulation of the brain state by confidence: Pupil-linked arousal

The state of brain networks is also modulated by neuromodulation, in particular the arousal state that is reflected in changes in pupil diameter. Subjects were fixating and the task was auditory, without visual stimuli or change in luminance, the pupil diameter therefore reflected here the subject's internal state. We distinguished two aspects of the pupil diameter: phasic and tonic levels, which we measured respectively as changes relative to a pre-stimulus baseline (-250 to 0 ms) and absolute changes. The phasic responses showed a transient increase in response to the stimulus (**Fig 4A**); we used a multiple linear regression for each peri-stimulus time point, including the Bayes-optimal surprise and Bayes-optimal confidence. The phasic levels showed only an effect of Bayes-optimal surprise (**Fig 4B**) whereas the tonic levels showed only an effect of Bayes optimal confidence: lower confidence was associated with larger tonic pupil size (increased arousal), see **Fig 4C** (cluster-forming p<0.05; cluster-level p<0.001, two-tailed, n = 10,000). The confidence effect on tonic pupil size was stable across peri-stimulus times, owing to the strong autocorrelation of this signal (autocorrelation for a 2 s lag, $\rho$ = 0.75 ± 0.13, median and SD across recording sessions).

## Pupil-linked arousal: Control analyses

In order to test the robustness of the confidence effect on tonic pupil size, we repeated the same multiple regression analysis but considering now the residual Bayes-optimal confidence (cluster-level p = 0.005, n = 10,000) and to control for the temporal profile of Bayes-optimal

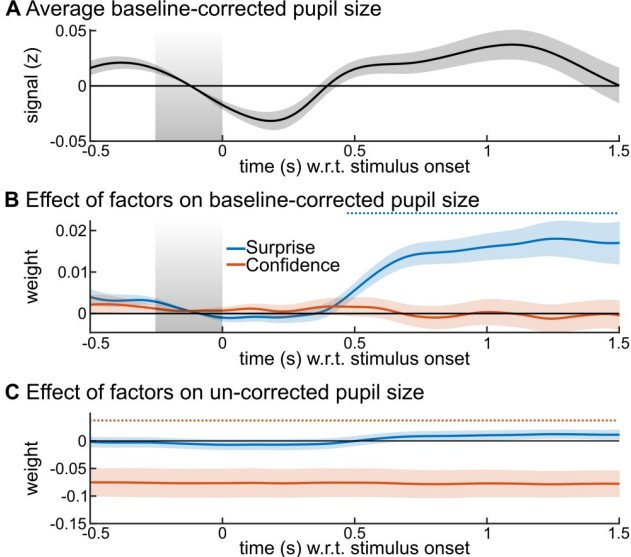

**Fig 4. Tonic and phasic pupil size distinctly reflect confidence and surprise.** The z-scored pupil signal was analyzed with mass-univariate regression. The value at each peri-stimulus time was regressed, across trials, using a multiple linear model with two predictors: the Bayes-optimal confidence about the probability of the current stimulus, the Bayes-optimal surprise. We used the baseline (see grey area) corrected pupil size as a measure of phasic response. (A) shows its average values and (B) the effects of predictors. We used the uncorrected pupil size as a measure of its tonic level; (C) shows the effect of predictors. Time-series show mean ± s.e.m.; the horizontal dashed lines indicate significant time points, corrected for multiple comparisons (cluster-forming p<0.05; cluster-level p<0.05, two-tailed, n = 10,000).

confidence, we used the same permutation analysis as described above (cluster-level p = 0.03 one-tailed, n = 10,000).

## Modulation of evoked responses by the brain state

The above analyses showed that the confidence about predictions modulates the brain states, in particular large-scale beta-range oscillations and the arousal state indexed by tonic pupil size. If those brain states play a role in the confidence-weighting of surprise, and thus perhaps the updating process, their level should modulate the confidence-weighted surprise responses evidenced in **Fig 2**, around 200 ms. In particular, they may capture trial-by-trial variations in the subject-specific states and thus account for the dampening of this surprise response on top of the effect predicted by the Bayes-optimal confidence. In order to test this possibility, we estimated two multiple regression models onto the evoked responses shown in **Fig 2B**, including the Bayes-optimal surprise, Bayes-optimal (residual) confidence, and the trial-by-trial value of either the post-stimulus beta-band activity (using the cluster shown in **Fig 2A**) or the tonic pupil size (averaged within -250 to 0 ms relative to stimulus onset). Bins of successive trials were used to achieve more robust results, see Methods. We corrected results for multiple comparisons using a cluster-forming p<0.05 and cluster-level p<0.05, one-tailed, n = 10,000 (**Fig 5**).

This analysis replicates the result found in Fig 2: an effect of Bayes-optimal surprise around 200 ms (as well as later) together with an opposite effect of Bayes-optimal confidence. More beta-band power being correlated with higher confidence, beta-band power should dampen the surprise response, whose sign was negative (corresponding to an inward field), this negative effect (dampening) onto a negative surprise response (inward field) should result in a positive regression coefficient, a result that we observed and which was significant only around 200 ms. By contrast, higher tonic pupil levels correlating with lower confidence, tonic pupil should therefore strengthen the surprise response, which was negative (inward field), an effect that should result in a negative regression coefficient, which we observed significantly and selectively around 200 ms.

In order to better interpret those effects, it is important to know whether those beta-band oscillations and the tonic pupil size are two sides of the same coin (but note that pupil data is available only in 18/24 subjects, reducing statistical power). If so, one would expect a negative trial-to-trial correlation, since those signals co-vary, respectively, positively and negatively with Bayes-optimal confidence. However, this correlation was negligible, if not even slightly positive (Pearson $\rho$ = 0.043 ± 0.03 s.e.m., 95% CI = [-0.02 0.11], Cohen's d = 0.36, $t_{17}$ = 1.53, p = 0.14). Instead of using separate multiple regression models, one can include those two signals in the same model of the evoked response, along with the Bayes-optimal surprise and confidence. The regression coefficients become difficult to interpret since the predictors (pupil size, beta-band power) share correlation with confidence, which itself correlates with the evoked response. However, we found a significant effect of pupil size from 115 to 220 ms (cluster-forming p<0.05 and cluster-level p<0.05, one-tailed, n = 10,000).

## Discussion

Subjects estimated the probabilities that characterize a sequence of inputs, and the confidence associated with those estimates, following (to some extent) the principles of Bayesian inference. Those estimates were accurate despite the presence of volatility. Bayesian inference leverages the confidence-weighting principle to remain accurate in the face of volatility, by dynamically modulating the update prompted by surprising observations. The MEG responses evoked by stimuli showed numerous signatures of surprise (vigorous responses to unexpected

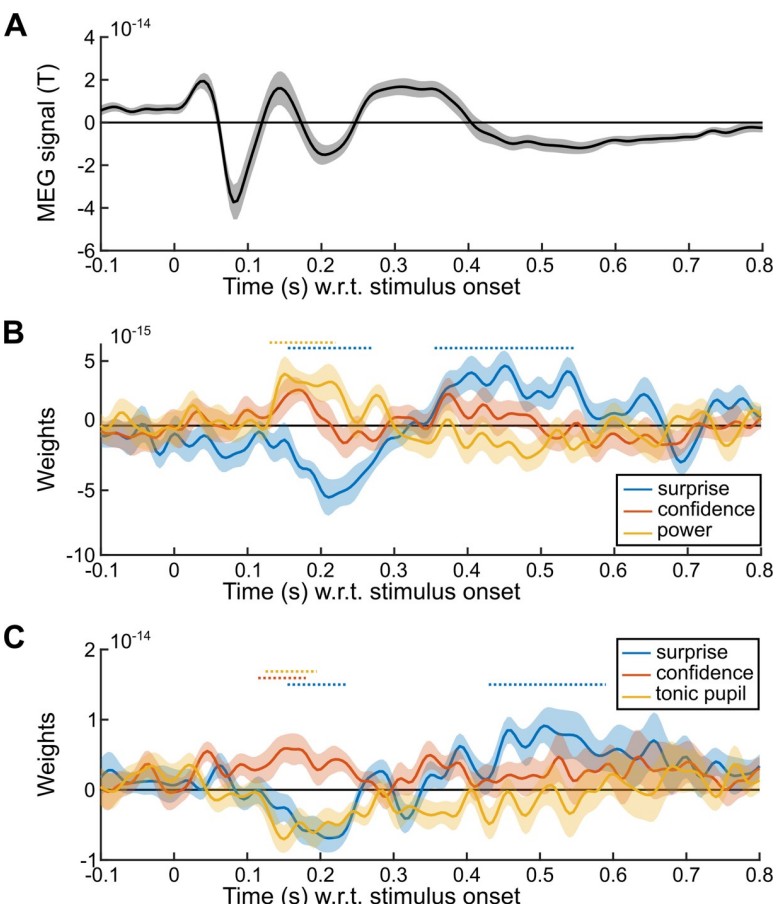

**Fig 5. Tonic pupil size and beta-range oscillations shape the evoked response.** The signal in the confidence cluster shown in Fig 2B was analyzed with mass-univariate regression. (A) Average MEG signal in the confidence cluster shown in Fig 2B. The sign of the signal is useful to interpret the effect of predictors. (B) Results of a multiple regression model comprising: the (residual) Bayes-optimal confidence about the probability of the current stimulus, the Bayes-optimal surprise and the power in the post-stimulus, beta band cluster (see Fig 3A). (C) Results of the same multiple regression model but with tonic pupil size instead of beta-band power. Time-series show mean ± s.e.m., horizontal dashed lines indicate significance corrected for multiple comparisons (cluster-forming $p<0.05$; cluster-level $p<0.05$, one-tailed, n = 10,000).

stimuli) and for one of them, an additional effect of confidence, conforming to the confidence-weighting principle. Whole-brain beta-band power increased with optimal confidence and was predictive of the actual subject's confidence. Lack of confidence increased the tonic pupil size while phasic dilation reflected surprise. Lower beta-band power and higher tonic pupil size increased the evoked, confidence-weighted surprise response. Overall, the results indicate that changes in brain state indexed by beta-band power and pupil-linked arousal, are related to differential responses to surprising stimuli, providing a mechanism for a dynamic confidence-weighting of learning.

We now discuss those results in the context of previous studies, which requires to delineate precisely what aspects of confidence we investigated here. It is the confidence that accompanies probability estimates on a trial-by-trial basis, thus it is not "global" or related to self-confidence [58]. It is also not related to whether the stimulus is perceived clearly or ambiguously, which is important in general for inference [12,18,59–61] but does not play a role here since stimuli were perceived without ambiguity. In addition, most results here are based on Bayes-

optimal confidence: the one that the subject *should* hold. It correlated with subject's confidence and beta-band power (which itself was also predictive of subject's confidence). Yet, Bayes-optimal confidence remains only an imperfect model of subject's confidence and both quantities, although related here, should not be conflated. Future work should investigate the algorithm (which the Bayes optimal model is not), i.e. the chain of computations that allows the subject to convert a given sequence of observations into a probability estimate and the associated confidence level. The present results may be useful to this end, by listing some properties that this algorithm should exhibit (confidence-weighting, several normative properties of confidence).

The type of confidence studied here (about a learned estimate) is also different from most studies, which focused on confidence about memory [62,63] and confidence about decisions [18,60,64]. Those different types of confidence correspond theoretically to different constructs: confidence about a memory or decisions correspond to the probability of the decision or memory being correct [60,65], whereas confidence about a learned estimate (even when this estimate is itself a probability) corresponds to a higher-order quantity: the precision of a posterior distribution [15,16,31,66].

Last, previous studies have not always disentangled confidence from related quantities. In many situations, confidence about predictions (i.e. posterior precision) is related to the expected outcome uncertainty (also referred to as *predictability* [67–69]). Our own analysis shows that confidence indeed increases, in the Bayes-optimal model and subjects, when the next outcome is more certain (probability closer to 0 or 1). Previous studies showed that the precision of predictions modulates reaction times [26], functional magnetic resonance imaging (fMRI) activity in fronto-parietal networks [70], pupil size [71], or MEG/EEG responses like the mismatch negativity [67–69,72], but with tasks and analyses that leave unclear whether the effect is genuinely about precision, or also about expected outcome uncertainty. A similar and related confound exists between expectation and attention, which are often related in practice but distinct in theory [39,67,68]. Here, we controlled specifically for this correlation by using the *residual* confidence in our analyses. This is important because many brain signals are associated with expected outcome uncertainty [28,39,67–71,73–75].

We now discuss more broadly the putative mechanisms of confidence-weighting in a learning context. Our interpretation is that confidence modulates the state of brain networks, which then process the feedforward input differently, in particular by increasing (low confidence) or decreasing (high confidence) the response to surprising stimuli, thereby modulating the effect of surprise onto updating. This interpretation is in line with the notion of neural gain: when the neural gain is higher (low confidence), expected and unexpected stimuli, for the same difference in expectation, elicit a larger difference in response. The neural gain is largely thought to depend on attention, neuromodulation and oscillatory activity in brain networks, three aspects that we discuss below.

The role of confidence on update described here is computationally identical to the one ascribed to selective attention: some observations are given more weight than others; this link has been already discussed by others [25]. This is also related to the notion of neural gain which enhances or suppresses the effect of a given stimulus on further, downstream processing [76]. Noradrenaline was proposed as a major modulator of the neural gain [55]. This is particularly interesting since we found modulations of tonic pupil size by confidence. Specifically, we found larger pupil size for lower confidence, which is consistent with previous studies [77]. Non-luminance based changes in pupil size reflect the arousal state and in particular, noradrenaline release. Correlation between pupil size and firing activity in the locus coeruleus, the main nucleus releasing noradrenaline in the brain [78], was found in both rodents [43] and macaque monkeys [79]. In addition, noradrenergic activity, unlike cholinergic activity, correlates with both fast and slow changes in pupil size [80]. The effect of confidence on tonic pupil

size found here could therefore arise from a change in noradrenergic activity. This noradrenergic activity changes neurons' membrane potential [81] and its slow fluctuations [43], promoting the selectivity of sensory processing, akin to the neural gain model. Similarly, activation of the locus coeruleus (and increased pupil size) promotes feature selectivity in the sensory domain [56,82]. Changes in pupil size are also related to changes in the processing of sensory information and performance during perceptual decision making tasks [83,84], with different effects of phasic and tonic pupil size [85] as in the present results.

In line with this noradrenergic modulation of neural gain, previous studies showed that larger tonic pupil size renders fMRI responses more extreme, i.e. lower or higher [76]. Larger pupil size and higher activity of the locus coeruleus are also associated with better memorization [86,87], and thus a long lasting effect of the current observations. Rodent studies showed that one-shot learning can occur with noradrenergic input to the hippocampus [88]. More generally, a role for noradrenaline in confidence-weighted learning is consistent with the fact that increased pupil size co-occurs with increased updating [36] and reset of current strategies [89], and that more volatile learning contexts are associated with larger pupil size [71,90]. Pharmacological studies in humans also provide supportive evidence, showing that manipulation of noradrenergic activity impacts the estimation of confidence in a decision task [91], learning dynamics when unexpected changes occur [92] and the effect of surprise on update [32,93].

The modulation of responses to new observations (notably surprising ones) by the current state of neural networks has also been associated repeatedly to specific rhythms of network activity, which provide signatures of networks' dynamics and computations [54,57]. This state-dependent modulation should aim at prioritizing new evidence when confidence is low, and on the contrary, preserve current estimates from conflicting evidence when confidence is high. Our data indicate that beta (and perhaps alpha) band power could play a key role in such a prioritization. This result is in line with the previous proposal that higher beta-band power preserves the current state of networks, promoting a "status quo" rather than a change [46] which may also be true for lower (alpha) frequencies [50]. Beta-band power in particular is associated with feedback signals [94], stronger attention and top-down control [95,96], and the prioritization of the current brain states over new inputs [47]. In support to this view, stronger low frequency (<30 Hz) oscillations attenuate early evoked sensory responses [52], increase the criterion of perceptual detection (resulting in reduced detection) by modulating baseline excitability [97–100] and degrade performance in perceptual decisions [101].

We acknowledge that the effect of confidence on the evoked response we report is weaker in strength, temporal and spatial extent, than the other effects (e.g. stimulus identity, surprise), which is certainly a weakness of our study. However, it is noticeable that the modulation of evoked surprise responses by confidence was confined to an early post-stimulus latency (around 200 ms) rather than occurring later, as could be expected for instance from the proposal that later brain waves like the P300 correspond to the updating, which in theory is confidence-weighted, and enhanced by attention [38,67,73,102–108]. However, those later brainwaves, in particular the P300, are not systematically a signature of update, for instance in a recent EEG study [33], the P300 was modulated by surprise, but equally and irrespective of the need to update the current estimate. In line with our result, another EEG study showed that the difference between expected and unexpected sounds (standard and deviant in an oddball task) was also larger around 175–200 ms (and after 350 ms) when pupil size was larger [109]. In rodents, activation of the locus coeruleus also increased the amplitude of multi-unit activity (in the sensory cortex involved in the task) specifically within 125–200 ms post-stimulus [44]. The mismatch negativity, a surprise response peaking around 170 ms, also seems better explained (in a roving paradigm) by confidence-weighted updates than by change detection,

adaptation or simple prediction errors [110]. Those studies are thus consistent with the latency of the confidence-weighted surprise response we found here. Source reconstruction showed that this effect may arise from the inferior frontal sulcus (among other regions), which is consistent with the location of the confidence-weighted surprise responses detected with fMRI in the same probability learning task as used here [19].

Subjects were fully aware of the stimuli in our task, and actively engaged in the estimation of the generative probabilities; thus, with the current study alone, one cannot know whether the same neural correlates of confidence (evoked responses around 200 ms, beta-band oscillations, tonic pupil size) would have been found had the subjects been ignorant about the task structure, or performing a task orthogonal to the estimation of probabilities, or even not paying attention to the stimuli.

Another limitation of the present study is that we do not demonstrate directly a link between confidence-weighted surprise responses and the updating of an internal model. This is because, in the present design, behavioral reports are occasional, making it impossible to measure how much the subject updates her prediction from one trial to the next. Future studies could use a denser sampling of behavior so as to test whether the size of the confidence-weighted surprise response (probed neurally) on a given trial is predictive of the ensuing update (probed behaviorally). All analyses are also based on correlations, which indicate the existence of associations but remain insufficient to interpret those associations as causal links.

Together, the studies quoted above and our own results support the idea that higher synchronization in alpha/beta band frequencies and lower noradrenergic activity could be mechanisms by which the brain shields current estimates against update prompted by surprising observations, thereby implementing a confidence-weighting mechanism during learning.

## Methods

### Participants

Twenty-four participants (16 women) aged between 20 and 34 (mean: 25.4, SD: 3.7) were recruited by public advertisement. They gave their written inform consent prior to participating; the study was approved by the Ethics Committee Ile de France VII (CPP 08–021). For one subject, one of the four sessions was unavailable due to a technical problem. Due to technical errors in saving files, the pupil diameter data is available only for 18 of the 24 subjects; analyses involving pupil size are restricted to those 18 subjects; all other analyses are performed on the 24 subjects.

### Task

This task and minor variants have been used previously [8,15,19]. The task was run using Matlab and Psychtoolbox. The experiment was divided into one training session, performed outside the MEG room, and four sessions in the MEG (lasting approximately one hour and a half with pauses). Each session presented a sequence of 380 auditory tones (one every 1.4 s), denoted A and B, corresponding to chords (350, 700 and 1400 Hz vs. 500, 1000 and 2000 Hz) that were perceived without ambiguity. Tones were presented in both ears, lasted 50 ms including 7 ms rise and fall times. The sequences were generated based on transition probabilities between successive tones; those probabilities were fixed only for a limited period of time delineated by change points. The change point probability was fixed on each trial, equal to 1/75; when a change point occurred, both transition probabilities were re-sampled within the interval 0.1–0.9, with the constraint that the change should be at least 4-fold for one of the probabilities, i.e. $|\log\left(\frac{p_{t-1}}{1-p_{t-1}}\frac{1-p_t}{p_t}\right)| > \log(4)$ for p = p(A|A) or p = p(B|B). The sequence was

occasionally interrupted by questions (median: every 13 trials, SD: 4.4), asking the subject to predict the identity of the next tone with a continuous slider (we reminded them about the identity of the previous tone, which is useful to report transition probabilities), and then the confidence about this prediction (slider whose ends were labeled "not at all" and "fully").

Subjects were thoroughly instructed about this generative process. In order to acquaint subjects with the notion of randomness and transition probabilities, we used a graphical display with animated "wheels of fortune". Each wheel corresponded to a pie chart, whose sections indicated the probability to repeat the same tone, or to change it, and each of the two charts corresponded to tones A or B, thus effectively representing transition probabilities. A ball rolled around the wheel with decaying speed, ending at a random position (in section "repeat" or "change") that triggered the onset of the corresponding tone. Subjects rolled the ball multiple times to generate short sequences. In order to explain the notion of change point, we introduced a key which, when pressed, changed the size of sections "repeat" and "change" randomly, in both wheels. Subjects then generated sequences with change points. We then explained to subjects that, during the task, they would only hear the sequences of tones and that they would have to figure out the underlying pie charts (i.e. transition probabilities) and the moment of change points. They performed at least one full session as training.

## Bayes-optimal learning model

The Bayes-optimal model used here is described elsewhere [8,15,111] and the corresponding Matlab code is available online: https://github.com/florentmeyniel/MinimalTransitionProbsModel. We used it with the following options: the learned quantities are "transitions", the estimation type is "HMM", the probability grid used for numeric integration has 20*20 values and the prior about transition probabilities is flat. Below, we summarize the main aspects of this model.

The model uses Bayes rule to infer optimally the posterior distribution of transition probabilities at any given trial, denoted $\theta_t$ (which is a pair of transition probabilities) given a set of assumptions $M$ and previous observations $y_{1:t}$:

$$p(\theta_t|y_{1:t}, M) \propto p(y_{1:t}|\theta_t, M)p(\theta_t, M) \tag{1}$$

Those assumptions closely correspond to the actual generative process: the change point probability $p_c = 1/75$ is given, and the model assumes that when a change point occurs, both transition probabilities are simultaneously re-sampled uniformly in the range 0 to 1 (it was not aware of additional constraints). The position of change points being unknown, the possibility that they may occur at any given trial must be taken into account. The generative process obeys the so-called Markov property: if one knows $\theta$ at time $t$, then the next observation $y_{t+1}$ is generated with $\theta_{t+1} = \theta_t$ if no change occurred and with another value drawn from the prior distribution otherwise. Therefore, if one knows $\theta_t$, previous observations are not needed to estimate $\theta_{t+1}$. The generative process can thus be cast as a Hidden Markov Model (HMM), which enables to iterate the computation of the joint distribution of $\theta$ and observations, starting from the prior, and updating this distribution by moving forward in the sequence of observations:

$$p(\theta_{t+1}, y_{1:t+1}) = p(y_{t+1}|\theta_{t+1}, y_t) \int p(\theta_t, y_{1:t})p(\theta_{t+1}|\theta_t)d\theta_t$$
$$= p(y_{t+1}|\theta_{t+1}, y_t) \left[ (1 - p_c)p(y_{1:t}, \theta_t = \theta_{t+1}) + p_c \int_{\theta_t \neq \theta_{t+1}} p(y_{1:t}, \theta_t)d\theta_t \right] \tag{2}$$

We computed this integral numerically by discretization on a grid. We obtained the posterior probability by normalizing this joint distribution.

## Formalization of prediction, surprise and confidence

The prediction, i.e. the probability of the next stimulus (question #1 asked to subjects) was computed from the posterior using Bayes rule. It is the mean of the posterior distribution of the relevant transition probability (the one that corresponds to the tone presented on the previous trial), which we note $\theta^{rel}$. Let $\theta^{rel}$ be probability of the next stimulus being A (not B), then:

$$
\begin{aligned}
p(y_{t+1} = A|y_{1:t}) &= \int p(y_{t+1}|\theta_{t+1}, y_t)p(\theta_{t+1}|y_{1:t})d\theta_{t+1} \\
&= E[\theta_t^{rel}|y_{1:t}]
\end{aligned}
\tag{3}
$$

The surprise corresponding to the actual new observation was defined, following [27], in bits, as the negative logarithm (to base 2) of the observation likelihood:

$$
surprise = -\log_2\big(p(y_{t+1}|y_{1:t})\big)
\tag{4}
$$

The confidence about the probability estimate (question #2) was computed as the log-precision of the posterior distribution of the relevant transition probability [15,16,18,31,66]:

$$
confidence = -\log(var[\theta_t^{rel}|y_{1:t}])
\tag{5}
$$

## MEG recording and pre-processing

We recorded brain activity using a whole-head MEG system (Neuromag Elekta LTD, Helsinki), sampled at 1 kHz with hardware bandpass filtering between 0.1 and 330 Hz. This systems has 102 triplets of sensors (1 magnetometer and two orthogonal planar gradiometers); we report results for the magnetometers only, but we used all sensor types for source reconstruction. We digitized the head shape (FASTRAK, Polhemus) including the nasion, pre-auricular points and various points on the scalp, and we measured head position within the MEG system at the beginning of each session with four head position indicator coils placed on the subject's head. Electro-oculograms, electrocardiogram and pupilometry were recorded simultaneously with MEG.

Raw MEG signals were first pre-processed with the constructor software MaxFilter, in order to correct for between-session head movement (recordings realigned on the first session), removing nearby magnetic interference and correcting for noisy sensors by means of Source Space Separation [112].

The data was further pre-processed with FieldTrip [113]. The signal was epoched from -1.4 s to 1.4 s relative to stimulus onsets, and -1.2 s to 0.5 s relative to question onsets. Epochs were visually inspected to reject those with abrupt jumps, spikes or muscular artifacts, and line filtered (50, 100, 150 Hz). Epochs were then decomposed with ICA; components were visually inspected and those resembling artifacts corresponding to eye blinks and heart beats were removed. Outlier epochs were rejected based on the signal variance and kurtosis. The mean number of trials finally included was 1335.8 ± 127 SD for stimuli and 90.2 ± 7.9 SD for questions. In order to analyze evoked responses, the signals were low-passed filtered (30 Hz) and down-sampled to 200 Hz. In order to analyze the spectral components, the signals was decomposed into peri-stimulus times (every 50 ms) and frequencies (every frequency between 7 and 40 Hz) with Morlet wavelets (using a $f/\sigma_f$ ratio of 7, where $f$ is the frequency and $\sigma_f$ is the spectral standard deviation of the wavelet).

No baseline correction was applied since trials are not independent from one another in our transition probability task.

## Source reconstruction

We acquired anatomical T1-weighted magnetic resonance images (3T Prisma Siemens scanner) for 18 of our 24 subjects with 1 mm resolution. This anatomical image was segmented to extract the cortical surface and head shape with FreeSurfer [114,115] and segmented tissues were imported into BrainStorm [116] to perform source reconstruction. For subjects without a personalized anatomical image, we used BrainStorm's default template (ICBM152). MEG and MRI data were co-registered using the digitized anatomical markers. All cortical meshes comprised ~15 000 vertices.

For stimulus-evoked responses, we estimated at the subject-level the sources corresponding to the signal averaged within 155–230 ms (relative to stimulus onset) and across trials with high and low Bayes-optimal surprise, and high and low Bayes-optimal confidence (using median split for both). We estimated the noise covariance matrix from the signal within -0.3 to -0.1 s relative to stimulus onset, concatenated across trials. We used a normalized minimum norm estimate of the current density map, with a loose orientation constraint orthogonal to the cortical sheet (parameter 0.2), corresponding to the option dSPM in BrainStorm. The subject-level differences of current's norm (high vs. low surprise, high vs. low confidence) where spatially normalized to the FSAverage atlas and analyzed statistically with a t-test at the group level. For display, the t-map was projected onto a high resolution mesh (~300,000 vertices).

For oscillations, we estimated at the subject-level the sources of the post-stimulus power in the beta band. Using FieldTrip, we computed the cross-spectral density for all sensor combinations, on each trial, using a tapper centered on 15 Hz with a smoothing window of ±3 Hz, and spanning 0.2–0.6 s after the stimulus presentation. We imported from Brainstorm into Field-Trip, the forward model and lead field model of each subject, which describe how activity in the source space translates into activity in the sensor space. We discretized the source space into a 3-dimensional grid with a 1 centimeter isotropic resolution. We then estimated, using all trials together, the beta power on each grid point using a beamformer (Dynamical Imaging of Coherent Sources), by computing a filter that projects sensor cross-spectral density data onto source space. This same filter was then used to project separately each condition (high vs. low confidence). We then computed the paired difference of power in source space, normalized the sources anatomically across subjects, and computed a group-level t-test for the condition difference. The resulting t-values were projected onto FieldTrip's default standard surface anatomy for visualization.

## Pupil size recording and pre-processing

Eye gaze and pupil size were monitored using EyeLink 1000. Blinks were delineated (adding a margin of 50 ms before and after) and the data within them linearly interpolated; the signal was then low-pass filtered (5 Hz) and epoched within -0.5 to 1.5 s relative to the stimulus onset. Epochs whose total blink duration exceeded 20% were excluded. Finally, the data was z-scored per subject.

## Regression models

All multiple regression models included a constant and z-scored predictors. The significance of regression coefficients were assessed with group-level t-tests. Regressions were estimated with Matlab's regstats.m function (based on QR decomposition), or custom code derived from it.

We often used the "residual" Bayes-optimal confidence as a predictor, which was computed by linearly regressing out several effects related to the prediction itself. More precisely, we estimated a multiple linear regression of the Bayes-optimal confidence, using as predictors the prediction itself (i.e. the Bayes-optimal estimate of the transition probability of the next stimulus, p(A)), its square and logarithm, log(p(A)) and log(p(B)). Using the square and logs enables to capture non-linear effects. The residual Bayes-optimal confidence was defined as the residuals of this multiple regression.

Most regression analyses of neural data include surprise and confidence, but not their interaction. This is because the model update, which we are ultimately interested in, consists in the present task mostly in an additive effect of surprise and confidence. We quantified this result as follows. We formalized the model update as the Kullback-Leibler divergence between the prior probability distribution (which is the posterior from the previous trial) and the current posterior, as in previous studies [19,29]. We performed two multiple linear regressions of the Bayes-optimal model update, both analyses included the Bayes-optimal surprise and confidence, with or without their interaction. The model without interaction resulted in a Pearson $\rho$ between fitted and actual update of 0.836, indicating that model update is to a large extent captured by the additive effect of surprise and confidence. Including also the interaction resulted in a negligible increase: $\rho = 0.839$. An alternative analysis would have been to regress neural data directly against the Bayes-optimal model update; however it is a weaker test for an update signal because the sole effect of either surprise or confidence would be sufficient to yield a significant regression.

The multiple regression of the evoked response shown in **Fig 5** involved the post-stimulus beta-band power, testing for a relation between the evoked response and power on the same trial. Note that the sensors and timing largely overlap; a spurious correlation is thus expected due to signal expression: when recorded MEG signals from this region are stronger or less noisy (e.g. due to artifacts, sensor noise or reduced brain-sensor distance related to breathing and motion) both the measured power and evoked response should be stronger. Luckily, our prediction is that when power is stronger, the evoked response should be weaker, due to the confidence-weighting mechanism, resulting in a sign opposite to the spurious (artifactual) correlation. We predicted that if the spurious correlation was driven by fast trial-to-trial variations in signal expression (compatible e.g. with breathing), averaging successive trials into bins would disrupt it. In contrast, the expected (non spurious) correlation between power and evoked response being driven by slower trial-to-trials changes in confidence and power, it should be more robust to this binning procedure; in practice the correlation between pairs of adjacent trials was 0.61 ($\pm 0.001$ s.e.m., $t_{23} = 65.8$, $p = 1.0 \ 10^{-27}$) for confidence, and 0.15 ($\pm 0.022$ s.e.m., $t_{23} = 6.85$, $p = 5.5 \ 10^{-7}$) for power (to be more conservative, we removed linearly the effect of Bayes-optimal confidence before computing this correlation). The effect of power onto the evoked response indeed greatly increased when using non-overlapping bins of 10 consecutive trials, specifically at the moment when such a correlation is expected (around 200 ms), see **S3A Fig**. This result holds for different bin sizes (**S3B Fig**). We reproduced qualitatively this effect with simulations (**S3C Fig**) under the following assumptions: 1) beta-band power linearly reflects Bayes-optimal confidence and has auto-correlated noise from trial-to-trial, 2) the (negative) evoked response linearly reflects surprise, beta-band power (negatively) and has non-correlated noise, 3) both signals are corrupted at the measurement level by the same non-correlated noise.

## Correction for multiple comparisons

For signals with two or more dimensions (times and sensors; times, sensors and frequencies), correction for multiple comparisons was computed with FieldTrip, following permutations and cluster-based statistics (sum of t-values in the cluster) effectively controlling the family-wise error rate [117]. Sensors located less than 4 cm apart were considered as neighbors (which corresponds to on average 3.9 neighbors per sensor). For signals with one dimension (time), we used a custom code implementing the same cluster-based, permutation test.

All analyses were initially corrected using cluster-forming and cluster-level alphas of 0.05, with two-tailed test (excepted when testing a specific direction of the effect, e.g. Fig 5). However, several analyses appeared significant with even more conservative thresholds, which may be more instructive to report as it advantageously limits the number and extent of clusters for interpretation and follow-up analyses; several analyses (Figs 2 and 3) are thus reported with more conservative thresholds. In the text, we systematically report the cluster-forming threshold as p-value, the cluster-level p-value and number of permutations used for the test (n).

## Cross-validated predictive accuracy

In order to estimate whether the subject's confidence could be predicted from the pre-stimulus power across different frequencies, we used a cross-validated ridge regression. The power was averaged within -0.5 to 0 s relative to the stimulus onset and z-scored across questions. We used a ridge penalty of 0.01, but the results are quite robust to the choice of this parameter. We adopted a cross-validation approach: the data were split into 20 distinct sub-sets of inter-leaved trials and at each iteration, the ridge regression was estimated on all sub-sets but one, and its parameter estimates were used to predict subjective confidence on the left-out sub-set. We assessed the accuracy of this prediction as the Pearson correlation between predicted and actual confidence at the subject's level.

## Supporting information

**S1 Fig. Temporal profile of confidence and surprise in the task.** We consider an example sequence in which observations are color-coded (blue and green). The top graph shows the posterior inference of transition probabilities made by the Bayes optimal model in the course of sequence presentation: the green dots show the probability for the next item to be blue if the previous item was green, the blue dots show the probability for the next item to be blue if the previous item was blue. The black line shows the prediction conditioned on the identity of the item previously presented. The middle graph shows the surprise (in bits) corresponding to the actual observations, whose identity on each trial is color coded; the sequence of colored dots therefore represents the sequence of observations. The bottom graph shows the Bayes optimal confidence (i.e. posterior precision) associated with the inferred transition probabilities, using the same convention as in the top graph. Several aspects are noteworthy. First, both surprise and confidence show marked dynamics within the course of an experimental session (380 stimuli). Second, those two dynamics are distinct, for instance, surprise may be rather steady while confidence changes (e.g. from stimulus 250 to 280). Third, predictions and the associated confidence can change repeatedly from trial-to-trial when transition probabilities differ, e.g. from stimulus 300 to 340. Fourth, similar predictions can be accompanied by different confidence levels (e.g. from stimulus 100 to 140, both transition probabilities support a prediction around 0.75 and yet, the prediction is associated with higher confidence when the previous observation is blue).
(TIF)

**S2 Fig. Post-stimulus beta-band power shows fast changes across trials and a gradual effect of confidence. (A)** Pairs of adjacent trials were sorted into high and low (red vs. blue) residual Bayes-optimal confidence on the current trial, and further sorted into high and low residual Bayes-optimal confidence on the next trial, therefore forming pairs that kept similar levels (dashed line) or changed drastically (plain line). The (z-scored) power in the post-stimulus beta-band cluster (Fig 3A) showed fast, trial-to-trial changes that parallel the residual Bayes-optimal confidence (*: $p < 0.005$, paired t-test). **(B)** The correlation between post-stimulus beta-band power and (residual) Bayes-optimal confidence was not driven by specific values but indeed corresponds to parametric changes. Error-bar: s.e.m.
(TIF)

**S3 Fig. Theoretical and empirical effect of binning on the multiple regression involving power vs. evoked responses. (A)**. The top panel is the same as Fig 5B. For this analysis, 10 consecutive trials were averaged into bins prior to estimating the regression. The bottom panel shows the results when trials are not binned (or equivalently, when there is only one trial per bin). Note the selective change around 200 ms for the effect of power. **(B)** explores this regression analysis specifically around 200 ms (averaging across significant time points, Fig 2A middle) and shows the significance (group-level t-value) of the effect of post-stimulus beta-band power onto to the ERF, depending on the number of consecutive trials per bin. **(C)** is the same analysis as in (B) but for signals simulated as follows: $POWER_{neural} = \beta_1{}^*CONF + \eta_{POWER}$, $ERF_{neural} = -(\beta_2{}^*SURP - \beta_3{}^*POWER_{neural}) + \varepsilon_{ERF}$, $POWER_{meas.} = POWER_{neural} + \varepsilon_{common}$; $ERF_{meas.} = ERF_{neural} + \varepsilon_{common}$. CONF and SURP are the Bayes-optimal surprise and confidence, $POWER_{neural}$ and $ERF_{neural}$ are the true beta-band power and evoked response, $POWER_{meas.}$ and $ERF_{meas.}$ are the signals measured by MEG; $\eta_{POWER}$ is auto-correlated Gaussian noise, $\varepsilon_{ERF}$ and $\varepsilon_{common}$ are identically distributed Gaussian noises. For the simulation, $\eta_{POWER}$ has SD = 1 and auto-correlation $\rho = 0.5$; $\varepsilon_{ERF}$ and $\varepsilon_{common}$ have SD = 1, $\rho = 0$; $\beta_1 = 0.25$; $\beta_2 = 0.25$, $\beta_3 = 0.5$. Those parameters are arbitrary, not fit to the data.
(TIF)

## Acknowledgments

I thank Micha Heilbron for contributing to set up the experiment; Micha Heilbron and Maxime Maheu for recording the entire dataset and Sébastien Marti for assistance during recording.

## Author Contributions

**Conceptualization:** Florent Meyniel.

**Data curation:** Florent Meyniel.

**Formal analysis:** Florent Meyniel.

**Funding acquisition:** Florent Meyniel.

**Investigation:** Florent Meyniel.

**Methodology:** Florent Meyniel.

**Project administration:** Florent Meyniel.

**Resources:** Florent Meyniel.

**Software:** Florent Meyniel.

**Supervision:** Florent Meyniel.

**Validation:** Florent Meyniel.

**Visualization:** Florent Meyniel.

**Writing – original draft:** Florent Meyniel.

**Writing – review & editing:** Florent Meyniel.

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
