## [Decision Letter · Decision Letter 0]

17 Feb 2020

Dear Dr. Meyniel,

Thank you very much for submitting your manuscript "Brain dynamics for confidence-weighted learning" for consideration at PLOS Computational Biology.

Sorry for the slow progress on this, partly due to difficulty in finding reviewers in the run up to Christmas.

As with all papers reviewed by the journal, your manuscript was reviewed by members of the editorial board and by several independent reviewers. In light of the reviews (below this email), we would like to invite the resubmission of a significantly-revised version that takes into account the reviewers' comments.

We cannot make any decision about publication until we have seen the revised manuscript and your response to the reviewers' comments. Your revised manuscript is also likely to be sent to reviewers for further evaluation.

Sincerely,

Jill O'Reilly

Associate Editor

PLOS Computational Biology

Samuel Gershman

Deputy Editor

PLOS Computational Biology

Reviewer's Responses to Questions

**Comments to the Authors:**

Reviewer #1: In this paper the author examines behavioural and neural correlates of optimal inference during a probabilistic learning task. In each trial, participants hear one out of two possible tones (A or B). In some trials participants are asked to predict the probability with which the tone will be repeated on the subsequent trial as well as their confidence associated to this probability estimate. During the experiment, the generative probabilities (p(A|A) and p(B|B)) can change at random points with very low probability. The task structure is transparent to the participants. Behaviour in the task approximates, to some extent, Bayes-optimal inference. Additionally, MEG data show signatures of optimal inference, in the sense that the neural activity in sensors representing surprise (or likelihood) is also modulated by bayesian confidence. This co-modulation of neural activity by the likelihood and confidence is consistent with Bayesian updating where likelihoods are upweighted when the variance of the prior is broad (i.e. low confidence). Further analyses show that brain state, indexed by pupil-related arousal and beta-band activity, covaries with confidence.

Overall the research presented in this paper is carefully done and thought of, following on the author’s previous work using the same probabilistic task and Bayesian framework. I have the following comments.

The behaviour of the participants in this paper is not analysed in depth. In Figure 1 D-E, some descriptive results are presented in relation to the predictions of the Bayes-Optimal model. Additionally, the probability estimates and confidence reports of the participants correlate, to some extent, with those of the bayesian model. These are the only points in which the relationship between the optimal model and behaviour is dealt with. I do not see these points as conferring evidence for or against the optimal model. In fact, the presented results are not informative. One for example could devise a simpler heuristic model, which could exhibit the same correlations with the bayesian estimates and confidence levels. The author is careful about this aspect, mentioning at several points that the bayesian model offers a formalisation of the learning process at the computational level. However, in light of the MEG data presented later, I am not sure if a computational level model —that presumably predicts broad features of behaviour but also misses on key algorithmic aspects of the learning process— is appropriate. Of course, if the optimal model was shown to explain the data better relative to other models the subsequent MEG results would bear larger importance. Below I provide a few recommendations on how to further scrutinise behaviour:

— It has been shown that people exhibit several biases when perceiving streams of data (e.g. hot hand fallacy, ignoring rare events etc) that are inconsistent with the optimal model. Are subjective probability and confidence estimates influenced by irrelevant features, for instance the fraction of alternations in the past 10 trials (keeping the counts for A and B fixed)?

— How well and how much faster (or slower) do people respond to change points relative to the optimal model?

— In intervals where the transition probabilities have larger entropies, do people report overall lower confidence?

— How correlated are reported probability estimates and reported confidence levels in the data and in the optimal model?

The above are just starting points towards falsifying (or failing to do so) the optimal model. I appreciate that in the absence of overt behaviour in most of the trials this point may be difficult to address. However, at least one simpler model should be compared both qualitatively and quantitatively to the optimal model.

2) The MEG analysis presented in Figure 2 shows that key quantities in the optimal model correlate with the evoked MEG activity. The validity of this result is directly dependent on the unresolved point above (how well does the optimal model describe the learning process). For instance, bayesian confidence and counter_A*counter_B (with counter_X keeping track of how many times X tone occurred in the last N trials in a suboptimal learning model) can be two strongly correlated quantities. Therefore the mere occurrence of these neural signals per se does not provide further evidence for the optimal model. The author here focuses on a subset of sensors on a specific time interval, which are jointly modulated by confidence and surprise, with opposite signs as prescribed by the optimal model. The rationale of this approach should be further motivated and validated. Wouldn’t it make also sense to show the activity of the “surprise” sensors as a function of high and low confidence? The encoding of confidence and surprise seems to be very variable: it appears lateralised (why?) and for the surprise metric there is a sign flip after 0.22 seconds, which I guess reflects the return of the signal back to baseline. All these aspects (the sign of encoding) appear to be arbitrary especially given the fact that overt behaviour is not used in any way to constraint this analysis (akin to the way beta band activity later on is used to predict subjective confidence; why is this not done for the sensor level data?).

3) The time-frequency analyses (Fig. 3) are presented as reflecting brain state fluctuations. It is not clear why for instance beta-band power here is related to arousal and is not evoked, related to task events. Additionally, in this section the authors speaks about arousal but does not clarify the direction of the relationship between confidence and arousal. It seems that higher confidence is associated with larger arousal (I inferred this indirectly). How can this effect be rationalised given also the fact that higher confidence is associated with evidence downweighting? Finally, the causality relation between arousal and confidence here is not determined. The author should discuss this limitation.

Minor comments:

— What does the residual bayesian confidence correspond to? I believe that if no change points could occur the bayesian confidence could be perfectly predicted by the regressors used here. However, it will be useful to gain an intuition of this residual confidence and what it captures.

— In Figure 2B statistics should be shown. Also there are four bins based (I believe) on independent median splits on confidence and surprise. It would be more informative if these bins where constructed in a conjunctive fashion (e.g. low confidence & low surprise).

— The analyses in Figure 5 is based on running two regression models. Given that the tonic pupil and beta power are not correlated why not run a single regression model?

— The writing is overall good but I had to resort to the author’s previous papers in order to clarify what each of the central metrics corresponds to (surprise, prediction, predictability etc). A longer introduction in which these key quantities are unpacked will be useful.

Reviewer #2: The general aim of Meyniel was to investigate whether confidence - formalized in terms of the precision of posterior predictive distributions of a Bayesian optimal model - modulates neural responses (MEG signal) evoked by surprising events during a learning task. A second and interrelated aim was to see whether confidence is associated with brain states (spectral power or pupil-linked arousal) and in turn, whether such confidence-weighted brain states are associated with surprise-evoked responses. The author found that confidence correlated with some evoked responses that overlapped in topography and time some responses associated with stimulus surprise (measured in terms of Shannon’s surprisal). Moreover, confidence was also associated with some brain states, namely, with tonic pupil size and with both pre- and post-stimulus onset alpha and beta power. These brain states were in turn associated with a surprise-evoked MEG signal response.

This is a very interesting and overall valuable study, and I think it could make an important contribution to contemporary literature on the predictive brain. To maximize its contribution, however, I think some conceptual and methodological issues should be addressed in a revision. I list these issues in the order in which they occurred to me.

Major comments:

1. The author was mainly interested in explore how confidence modulate surprise responses, an effect that is captured by their interaction. However, he states that their effect is theoretically mostly additive (lines 191-192). I think it is important to further argument this point: why is it theoretically mostly additive? Any reference? Why should the absence of an interaction not be considered a limit on the conclusion that surprise responses are dampened for higher confidence?

2. The effect of confidence on MEG evoked responses seems little and, indeed, it does not reach statistical significance in the analysis whose results are shown in Fig 5B. I think this aspect should be explicitly mentioned in the text since this effect is at the core of the first goal of the study.

3. Previous studies have shown that behavioral and neural responses associated with the amount of surprise of a stimulus can be distinguished from responses associated with how much the internal model is updated (e.g., O’Reilly et al. PNAS 2013; Kobayashi & Hsu, JNeurosci 2017; Visalli et al., NeuroImage 2019). As also reported in the discussion, the precision of the posterior after trial t is also linked with how large is the divergence with the posterior at trial t+1. In other words, the precision is inversely related with the amount of updating (often measured using the Kullback-Leibler divergence). I was wondering why the author did not regress out also this updating measure in computing the corrected confidence (the Matlab code provided online also calculates the DKL). Moreover, I think that it would be interesting to see whether the confidence-modulated brain states are associated with or modulate possible DKL responses.

4. Linked to the previous point, in the discussion the author states that “Overall, the results indicate that changes in brain states indexed by beta-range power and neuromodulation, are related to differential responses to surprising stimuli, providing a mechanism for a dynamic confidence-weighting of learning”. However such link with learning is indirect and not directly tested in the study. I think this fact should be addressed in the Discussion section.

5. Different critical alpha for both cluster formation and cluster-level correction were used from one analysis to another. Which was the reason? I think that such choices need to be justified.

Minor comments:

6. In the methods I would better formalize and describe the regression models using formulas for each model and reporting the employed software. Moreover at lines 577-581, it is reported: “We often used the “residual” Bayes-optimal confidence as a predictor, which was computed by linearly regressing out the effect of prediction and predictability. More precisely, we estimated a multiple linear regression of the Bayes-optimal confidence, using as predictors the prediction itself (i.e. the Bayes-optimal estimate of the transition probability of the next stimulus, p(A)), its square and logarithm, log(p(A)) and log(p(B))”. However, predictability was defined at line 128 as the entropy of the prediction. What do square and log of p(A) refer to?

7. At line 152, the term “factor” is used to refer not only to the categorical variable (stimulus identity), but also to the continuous variables. I would use a more appropriate term, such as “predictor” or “regressor” in all the manuscript.

8. At line 163, it is reported an effect of surprise between 80 and 155 ms, however, I thought that the analysis were run starting from 100 ms. Related to this point: why does the author not start the analysis from target onset? Which was the criterion to select the analysis time window. Why does the MEG signal analysis end at .8 s while the frequency analysis ends at .9 s?

9. Linked to comment 2, the topoplots in Fig 2 have a colormap in the range [-3.5, 3.5] t-value. However, the max observed t has a value of ~10. It would be better to use different colormaps for each effect (or the same color map in the range [-max|T| max |T|] to be able to perceive the difference in effect size between those effects (this applies also to Fig. 3).

10. Why in figs. 2B and C is surprise differently categorized in high and low?

11. At lines 207-208, the sentence “Power in low frequencies (<40 Hz) typically characterizes the state of large networks” needs references.

12. In the Discussion (lines 431-432), it is reported that “It is noticeable that the modulation of evoked surprise responses was confined to early post- stimulus latencies (around 200 ms) rather than occurring later”, however an effect of surprise was also found later.

13. There is a typo in the right subpanel (topoplot) of Fig. 2A: [0.9 0.14]s instead of [0.09 0.14]s

Reviewer #3: This manuscript reports data from an MEG study with healthy subjects who performed a perceptual learning auditory task, providing occasional behavioural responses pertaining to their sensory predictions and associated confidence. Pupil size was also monitored over time to test its relationship with confidence up-dating and surprise. The task used here is very close to the ones used by the author and collaborators in several previous studies. It is coupled here with Bayesian modelling and the analysis of various MEG components (evoked and oscillatory) at both the scalp and source level, to carefully assess the neural correlates of prediction, surprise and confidence, respectively. The main results that pertain to showing that the trial wise up-dating of confidence is well captured by Bayesian learning and can be related to evoked activity around 200ms, as well as beta power and pupil size, are of great interest to the field.

The manuscript is well structured and clearly written. Data analysis include several rigorous verifications that attest the robustness of the main findings. Although the manuscript would gain in readability if some of these additional analysis would be only shortly referred to in the main text and moved to an additional material section.

The discussion is quite thorough and put the present work in a useful perspective by relating it broadly to relevant recent studies pertaining to different domains (modelling, animal and human data, neurotransmitters and oscillations…).

This paper is suitable for publication and will be of interest to many. I only have minor suggestions which I hope will help further strengthening the manuscript.

General comments and questions

- How specific is this task? Subjects are indeed heavily trained and informed in details about the task hidden structure. Does it mean that the observed computations and the identified neural correlates are hardly generalizable?

- Importantly, the crucial and related but different notions manipulated here such as prediction, predictability, surprise, prediction error should be more clearly and initially defined. If some of these terms are interchangeable, they could be mentioned once but then one notion should be referred to by a single term. Especially as the author rightfully insists on the importance and the difficulty in distinguishing between such notions like precision and predictability.

- Regarding source reconstruction: please clarify whether you computed the source of the difference (source of an effect) or the difference of the sources (the former yields more sensitive results). And regarding the interpretation of the qualitative findings, the author pin-points the implication of the right IFG and intermediate precentral sulcus (for surprise), however the observed topographies reveal a more spread network including TPJ and other regions, please complement or nuance your current report. Along the same vein, the sources pertaining to confidence do not pin-point towards a clear region or network. Please comment. And in relation to that, could you try to complement this source reconstruction by source localizing the effect of confidence on post-stimulus beta power?

- why not reporting the effect (or absence of effect) of prediction onto low frequency oscillations and pupil size?

More specific and minor comments

- On figures 1.D, 1.E but also 3.D and S2.B, it is not clear to me how the data points were binned in order to build those figures

- lines 141 to 144: please unfold this argument which remains unclear to me.

- Fig. 2B reports MEG evoked signals for conditions obtained by splitting data around the median. Since such a split is performed along both confidence and surprise, could you please report the number of trials that fell into each of the four categories? Or clarify your strategy. Is it that after having split along one dimension, the data were further split along the other one? In which case it would be worth reporting the median values.

- I found the use of the loose term ‘Brain state’ a bit disturbing. I would encourage the author to envisage reconsidering that use.

- In the discussion, the part pertaining to related (MMN) findings using EEG, the author could also refer to:

Ostwald et al. Evidence for neural encoding of Bayesian surprise in human somatosensation. NeuroImage 62, 177–188 (2012).

Lecaignard et al. Implicit learning of predictable sound sequences modulates human brain responses at different levels of the auditory hierarchy. Frontiers in Human Neuroscience 9, (2015).

- Line 471: what you mean here is unclear to me.

- I noticed a couple of typos:

end of line 411: noradrenaline

line 500 (remove ‘a’: is given);

Fig. S2 line 654 (Paris OF adjacent…);

**Have all data underlying the figures and results presented in the manuscript been provided?**

Reviewer #1: No:

Reviewer #2: Yes

Reviewer #3: No: Part of the code is already available. The full raw data will be made available publicly upon publication.

PLOS authors have the option to publish the peer review history of their article (what does this mean?). If published, this will include your full peer review and any attached files.

Reviewer #1: No

Reviewer #2: No

Reviewer #3: No
---

## [Decision Letter · Decision Letter 1]

7 May 2020

Dear Dr. Meyniel,

We are pleased to inform you that your manuscript 'Brain dynamics for confidence-weighted learning' has been provisionally accepted for publication in PLOS Computational Biology.

Best regards,

Jill O'Reilly

Associate Editor

PLOS Computational Biology

Samuel Gershman

Deputy Editor

PLOS Computational Biology

Reviewer's Responses to Questions

**Comments to the Authors:**

Reviewer #1: All my comments have been addressed. I recommend acceptance of the paper.

Reviewer #2: The authors have addressed my comments. As a minor note, line numbers in the rebuttal did not match line numbers in the manuscript making it harder going through the revision

Reviewer #3: Thank you, all my comments have been addressed. This paper is now suitable for publication.

**Have all data underlying the figures and results presented in the manuscript been provided?**

Reviewer #1: No:

Reviewer #2: No: The author states that the raw data-set will be made available publicly upon publication.

Reviewer #3: Yes

PLOS authors have the option to publish the peer review history of their article (what does this mean?). If published, this will include your full peer review and any attached files.

Reviewer #1: No

Reviewer #2: No

Reviewer #3: No

---

## [Editor Report · Acceptance letter]

26 May 2020

PCOMPBIOL-D-19-02072R1 

Brain dynamics for confidence-weighted learning

Dear Dr Meyniel,

I am pleased to inform you that your manuscript has been formally accepted for publication in PLOS Computational Biology. Your manuscript is now with our production department and you will be notified of the publication date in due course.

With kind regards,

Laura Mallard
